# Health impacts and environmental footprints of diets that meet the Eatwell Guide recommendations: analyses of multiple UK studies

Pauline Scheelbeek [1,2] Rosemary Green,[1,2] Keren Papier,[3] Anika Knuppel,[3] Carmelia Alae-Carew,[2] Angela Balkwill,[3] Timothy J Key,[3] Valerie Beral,[3] Alan D Dangour[1,2]

¹Centre on Climate Change & Planetary Health, London School of Hygiene & Tropical Medicine, London, United Kingdom
²Department of Population Health, London School of Hygiene & Tropical Medicine, London, United Kingdom
³Nuffield Department of Population Health, Oxford University, Oxford, UK

**Correspondence to**
Dr Pauline Scheelbeek;
Pauline.Scheelbeek@lshtm.ac.uk

## ABSTRACT

**Objectives** To assess the health impacts and environmental consequences of adherence to national dietary recommendations (the Eatwell Guide (EWG)) in the UK.

**Design and setting** A secondary analysis of multiple observational studies in the UK.

**Participants** Adults from the European Prospective Investigation into Cancer - Oxford(EPIC-Oxford), UK Biobank and Million Women Study, and adults and children aged 5 and over from the National Diet and Nutrition Survey (NDNS). Primary and secondary outcome measures risk of total mortality from Cox proportional hazards regression models, total greenhouse gas emissions (GHGe) and blue water footprint (WF) associated with 'very low' (0–2 recommendations), 'low' (3–4 recommendations) or 'intermediate-to-high' (5–9 recommendations) adherence to EWG recommendations.

**Results** Less than 0.1% of the NDNS sample adhere to all nine EWG recommendations and 30.6% adhere to at least five recommendations. Compared with 'very low' adherence to EWG recommendations, 'intermediate-to-high adherence' was associated with a reduced risk of mortality (risk ratio (RR): 0.93; 99% CI: 0.90 to 0.97) and −1.6 kg $CO_2$eq/day (95% CI: −1.5 to −1.8), or 30% lower dietary GHGe. Dietary WFs were similar across EWG adherence groups. Of the individual Eatwell guidelines, adherence to the recommendation on fruit and vegetable consumption was associated with the largest reduction in total mortality risk: an RR of 0.90 (99% CI: 0.88 to 0.93). Increased adherence to the recommendation on red and processed meat consumption was associated with the largest decrease in environmental footprints (−1.48 kg $CO_2$eq/day, 95% CI: −1.79 to 1.18 for GHGe and −22.5 L/day, 95% CI: −22.7 to 22.3 for blue WF).

**Conclusions** The health and environmental benefits of greater adherence to EWG recommendations support increased government efforts to encourage improved diets in the UK that are essential for the health of people and the planet in the Anthropocene.

## BACKGROUND

Diets are likely to play a crucial role in the Anthropocene in supporting population

### Strengths and limitations of this study

► This is the first study (in a UK context) using empirical data to study health impacts and multiple environmental consequences of sustainable diets.
► The study uses multiple high-quality datasets with a total of 557 722 participants for health outcomes and 5747 participants for environmental footprints.
► The provided methods can be replicated in other settings and the Eatwell Guide dietary recommendations share many features of healthy lower environmental impact diets.
► Despite several sensitivity analyses, there might be residual confounding (ie, unmeasured differences between people who eat different diets) distorting our findings.
► Although the evidence base and quality of methods and metrics for environmental footprints are rapidly improving, uncertainty about the exact measurements of water footprints and greenhouse gas emissions of food items and diets in general remain.

health and safeguarding environmental sustainability for future generations. Current diets are associated with a high burden of disease: globally ~1.9 billion adults are overweight or obese, 462 million are underweight[1] and over 30% of the world's population suffers from deficiencies of essential nutrients.[2] The food system that produces these diets is also responsible for 21%–37% of global greenhouse gas emissions (GHGe)[3] and agriculture alone accounts for ~70% of fresh water withdrawal.[4] While food system GHGe contribute to global climate change problems regardless of location of production, food system water use is highly location specific: with approximately half of all countries classified as 'water scarce'—and a number of water safe countries projected to become water scarce by 2040[5]—origin of food supply is a crucial

consideration when considering the sustainability of food system water use.

There is an urgent need for significant transformations of the food system to produce diets that address both health and environmental concerns, and evidence on the recommended composition of these diets is expanding rapidly.[6 7] While the specific composition of such diets has been shown to vary considerably culturally and regionally compared with existing consumption patterns, these diets typically have substantially greater plant-based food content as well as no more than moderate content of animal sourced foods (eg, meat and dairy).[6 8 9]

The UK food system is no exception to these concerns for sustainability, and many transformative changes need to be made to make it more sustainable, resilient and healthy. Currently 64% of the adult population in the UK are overweight or obese,[10] and only 29% of adults and 18% of children between 5 and 15 years of age meet the recommended fruit and vegetable intake of '5 a day'.[10] At the same time, water use of UK diets is on average 2757 L/capita/day, which is below the global average of 3167 L,[11] but half of the national blue (surface and ground) water footprint (WF)—15.0 million m$^3$/day—is imported (ie, embedded in imported foods from elsewhere) from countries with water scarcity.[12 13] Furthermore, GHGe of average UK diets were found to be 1210 kg $CO_2$eq/capita/year as compared with an European Union average of 1070 kg $CO_2$eq/capita/year.[14] Evidence suggests that 17% of emissions could be avoided when the population were to shift to WHO dietary guidelines.[15]

Governments are increasingly including both health and environmental considerations in their recommended dietary guidelines. In the UK, Public Health England produced the 'Eatwell Guide' (EWG) as a 'policy tool to define government recommendations on eating healthily and achieving a balanced diet'.[16] From a health perspective, the EWG promotes, for example, cereals, potatoes, fruit, vegetable and fibre consumption, while recommending a limited consumption of sugar and processed meats.[17] Adhering to these individual guidelines has been associated with several health benefits including improved cardiovascular health[18] and reduced cancer risk.[19 20] From an environmental perspective, the EWG mentions the importance of a 'balance of healthier and more sustainable food', while providing information about protein alternatives, such as beans, peas and lentils, that typically have a lower environmental footprint than animal source food protein sources.[20–23] Compared with current diets, the EWG recommendations are, therefore, expected to have on average lower environmental footprints (GHGe, water and land use requirements).[24] The guidelines on meat and dairy, which are both set substantially below current average intake, were projected to have the largest impact on reduction of GHGe, land use and eutrophication.[25] GHGe of meat eaters in the UK was found to be roughly double that of vegans.[23]

While modelling studies have estimated the change in GHGe when shifting from current EWG adherent diets,

these are subject to many assumptions related to the substitutions between food groups inherent to the dietary change. To date, no study has been conducted using empirical dietary data (of large-scale cohort studies) to assess 'real-world' composition of diets that are in adherence with the EWG, which could substantially improve the uncertainty of estimation of the associated change in dietary GHGe. Furthermore, to date, no analysis of the WF of EWG adherence has been published.

In this report, we use high-quality data from three large prospective UK cohort studies to assess the health impacts associated with adherence to EWG dietary guidelines; and using nationally representative dietary intake data, we estimate the environmental footprints of UK diets with varying degrees of adherence to EGW recommendations.

## METHODS
### Datasets
We used four high-quality data sources in this paper (detailed description of each database provided in online supplementary appendix 1). The databases from European Prospective Investigation into Cancer - Oxford (EPIC-Ox),[26] UK Biobank (UKB)[27] and the Million Women Study (MWS)[28] contain comprehensive health information, linked death registration data as well as dietary intake data. These three datasets were used to estimate the associations with health of adherence to EWG recommendations. Details on the specific datasets have been published elsewhere. Briefly, participants in the MWS were recruited from women invited for breast cancer screening in England and Scotland between 1996 and 2001. Dietary intake was collected using semiquantitative questions and total mortality was determined using death records. We used data from 464 078 participants of the MWS database. In the EPIC-Ox study, which involves both male and female participants, dietary intake was collected using a Food Frequency questionnaire, while total mortality was ascertained using death record linkage. We used data from 40 030 men and women of the EPIC-Ox database. For the UKB study, middle-aged adults were recruited between 2006 and 2010. A subsample completed a minimum of three 24 hours dietary recall questionnaires. Participant data have been linked to the National Health Service (NHS) Central register to obtain mortality information. We used data from 53 614 participants of the UKB study. Finally, the National Diet and Nutrition Survey (NDNS)[29] contains nationally representative detailed dietary intake data that were used to analyse the diet-related environmental footprint of NDNS participants with different levels of EWG adherence. We excluded children<5 years of age from the NDNS data, as the EWG recommendations are not applicable to this age group.

### EWG dietary recommendations
Dietary intakes reported in each of the four databases were compared with recommended intakes by the EWG

and dichotomised (yes/no) to reflect individual adherence to EWG recommendations (recommendations by age and sex provided in online supplementary appendix 2). Nine food and nutrient groups with recommended levels of consumption specified in the EWG were considered: fruit and vegetables, oily fish, other fish, red and processed meat, total fibre, total salt, free sugars, saturated fatty acids and total fat. Two further EWG recommendations on protein and carbohydrates were excluded as significant heterogeneity across foodstuffs included in the questionnaires limit conversion from % of food energy intake to grams/day.[30] Participants were grouped into three categories of adherence based on the number of dietary recommendations met (total=9): very low adherence (score 0–2), low adherence (score 3–4) and intermediate-to-high adherence (score 5–9).

### Health impacts

We used multivariable-adjusted Cox proportional hazards regression models to assess associations between adherence to the EWG dietary guidelines and risk of total mortality, ascertained through death registries using participant data of EPIC-Ox, the MWS, and a subset of UKB with detailed dietary data. These estimates were combined using meta-analytical methods to provide pooled risk ratios (RRs). The mean follow-up time was 21.0 years in EPIC-Ox, 10.5 years in MWS and 3.9 years in UKB.

Participants in each database were excluded from the analysis sample if: (1) they had prevalent and/or unknown status of malignant cancer, diabetes or cardiovascular disease (data based on self-report and health record data) or rated their overall health as either poor or fair at recruitment; (2) they had energy intakes outside the ranges 2093–14 654 kJ for women and 3349–16 747 kJ for men, and did not report: a change in diet because of illness (MWS), not eating or drinking normally because of illness or fasting (UKB), because of stomach problems, bowel problems or diabetes (EPIC-Ox) and in UKB had not completed a minimum of three WebQ questionnaires (online dietary questionnaire); (3) they were lost to follow-up during the first 5 years of follow-up (MWS and EPIC-Ox only) and (4) their smoking status was unknown.

Associations were stratified by sex, region and method of recruitment (in addition to the general recruitment strategy, specific underrepresented groups were targeted for recruitment by leaflets—which could have introduced selection bias), where appropriate. All analyses were adjusted for smoking, deprivation, alcohol consumption, height, body mass index (BMI), exercise levels, hormone replacement therapy use, education, high blood pressure or hypertension and energy intake (see online supplementary appendix 3 for details). We performed a set of seven sensitivity analyses, comparing the above model with (a) an unadjusted model, models without adjustment for (b) energy, (c) height, (d) BMI or (e) smoking, (f) a model mutually adjusting for all other eight food

groups and (g) a model excluding smokers (see online supplementary appendix 4).

### Environmental footprints

We used data from NDNS waves 5–9 (2012–2017) to map the environmental footprints of diets in the UK. The database comprises detailed dietary data for 5747 individuals aged 5 years and over, grouped into 158 distinct food group aggregates. Data collection methods are described in detail elsewhere.[31] We used the Food and Agriculture Organization bilateral trade database to estimate the mean proportion of each food group imported from outside the UK.[32] The trade database includes bilateral data on exports and imports of all food and agricultural products reported by all the countries in the world.

#### Greenhouse gas emissions

Emissions of GHGs across the life cycle (kg $CO_2$eq/kg food) for the 158 distinct food group aggregates were derived from the published data (see online supplementary appendix 5 and 6). A weighted average of GHGe was calculated based on consumption of individual foods within each food group and proportion of supply from different countries. For foods entirely or more than 90% produced in the UK, UK-specific data were used. A weighted average for GHGe was applied for imported foods based on the proportion of total supply from various countries (see online supplementary appendix 7).

#### Water footprints

The blue (ground and surface water) WF (L/g food) of crop and livestock products were derived from the published data for 1996–2005 from the Water Footprint Network (WFN)[33] (see online supplementary appendix 5 and 6). For foods entirely or more than 90% produced in the UK, UK-specific WFN values were used. Imported food groups were assigned weights proportional to percentage of overall supply of each major exporting country to the UK, multiplied by WFN estimates for that particular country and food group (see online supplementary appendix 7).

The estimated GHGe and WFs associated with each food group were used to quantify total environmental footprints associated with the daily diet of each participant in the NDNS database. We compared GHGe and WFs of diets of those adhering and those not adhering to each EWG dietary guidelines, and estimated the mean change in environmental footprint that would occur if individuals shifted from low-to-intermediate/high adherence to the EWG guidelines.

## RESULTS

### EWG adherence

Less than 0.1% of the NDNS sample (0.078%) adhered to all nine EWG recommendations (figure 1A), with the largest proportion of the population (44%) adhering to 3–4 guidelines. The most commonly unmet

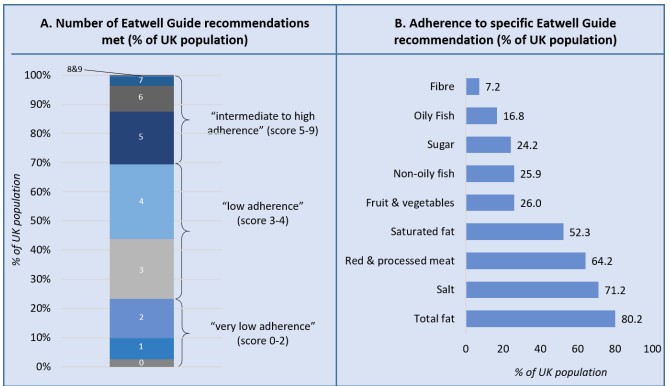

**Figure 1** Adherence to the Eatwell Guide recommendations by the UK population—based on data from wave 5–9 of the National Dietary and Nutrition Survey (NDNS). (A) Total number of recommendations met by % of UK population. (B) Adherence to specific recommendations.

recommendations included those on consumption of dietary fibre and oily fish (7.2% and 16.8% adherence, respectively), while more than 50% of the population met total and saturated fat, salt and red and processed meat recommendations (figure 1B). Adherence to the EWG recommendations in EPIC-Ox, MWS and UKB showed a

similar pattern to that in the NDNS data set (see online supplementary appendix 8).

## Health effects of adherence to EWG recommendations

Compared with those who had a very low adherence to the EWG, individuals with intermediate-to-high adherence had a 7% (99% CI: 3% to 10%) reduced risk of total mortality (figure 2). Sensitivity analysis identified smoking as an important confounder, and hence the main analysis was adjusted for smoking. Other potential confounders showed to only marginally affect associations detected in the main model.

Adherence to the recommendation on fruit and vegetable consumption was independently associated with the largest reduction in total mortality risk: a reduction of 10% (RR: 0.90; 99% CI: 0.88 to 0.93) (figure 3; attenuated to 9% in models adjusting for all other EWG recommendations see online supplementary appendix 4). Meeting the recommendations on saturated fat and oily fish consumption showed smaller associations with health benefits, with 5% and 3% reductions in mortality, respectively, (both attenuated to 3% in models adjusting for all other EWG recommendations see online supplementary appendix 4). There was no consistent evidence of an association with mortality risk for adherence

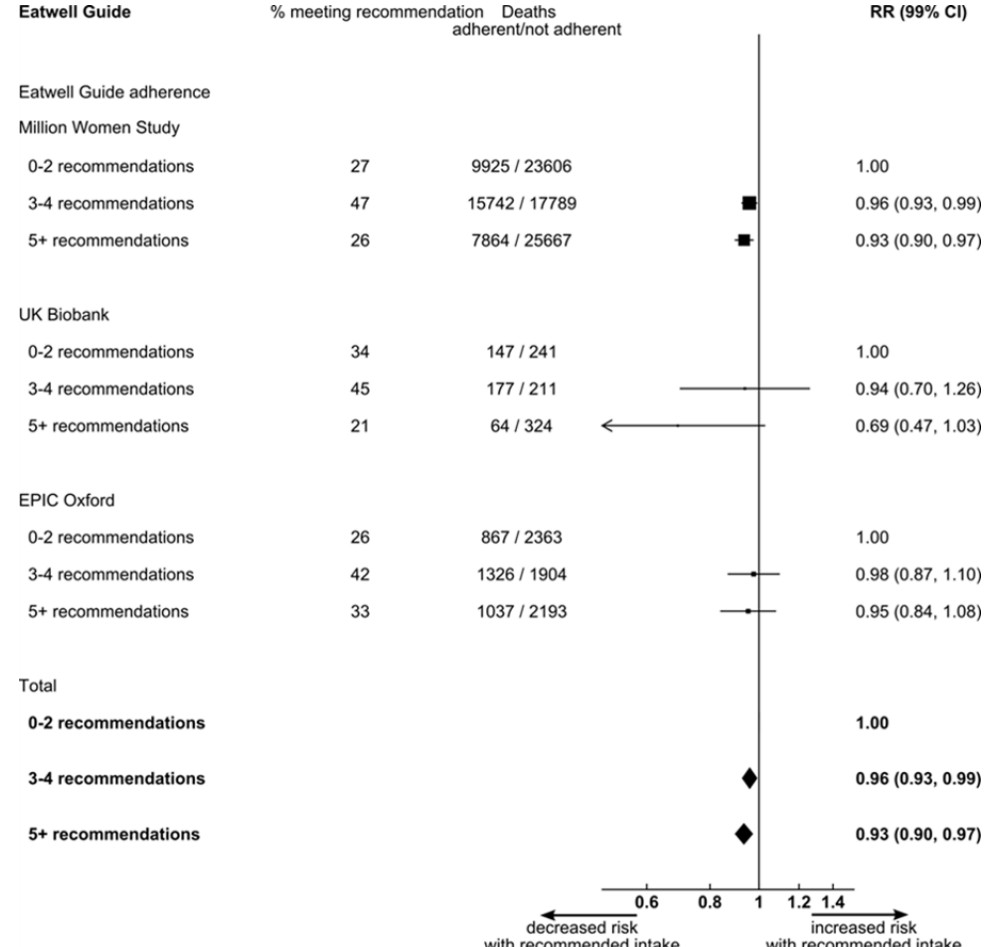

**Figure 2** Forest plot showing the study specific (Million Women Study, UK Biobank and European Prospective Investigation into Cancer - Oxford [EPIC Oxford]) and pooled mortality risk ratios comparing very poor adherence to Eatwell Guide rocmmendations (score 0–2) with poor adherence (score 3–4) and intermediate-to-high adherence (score 5–9).

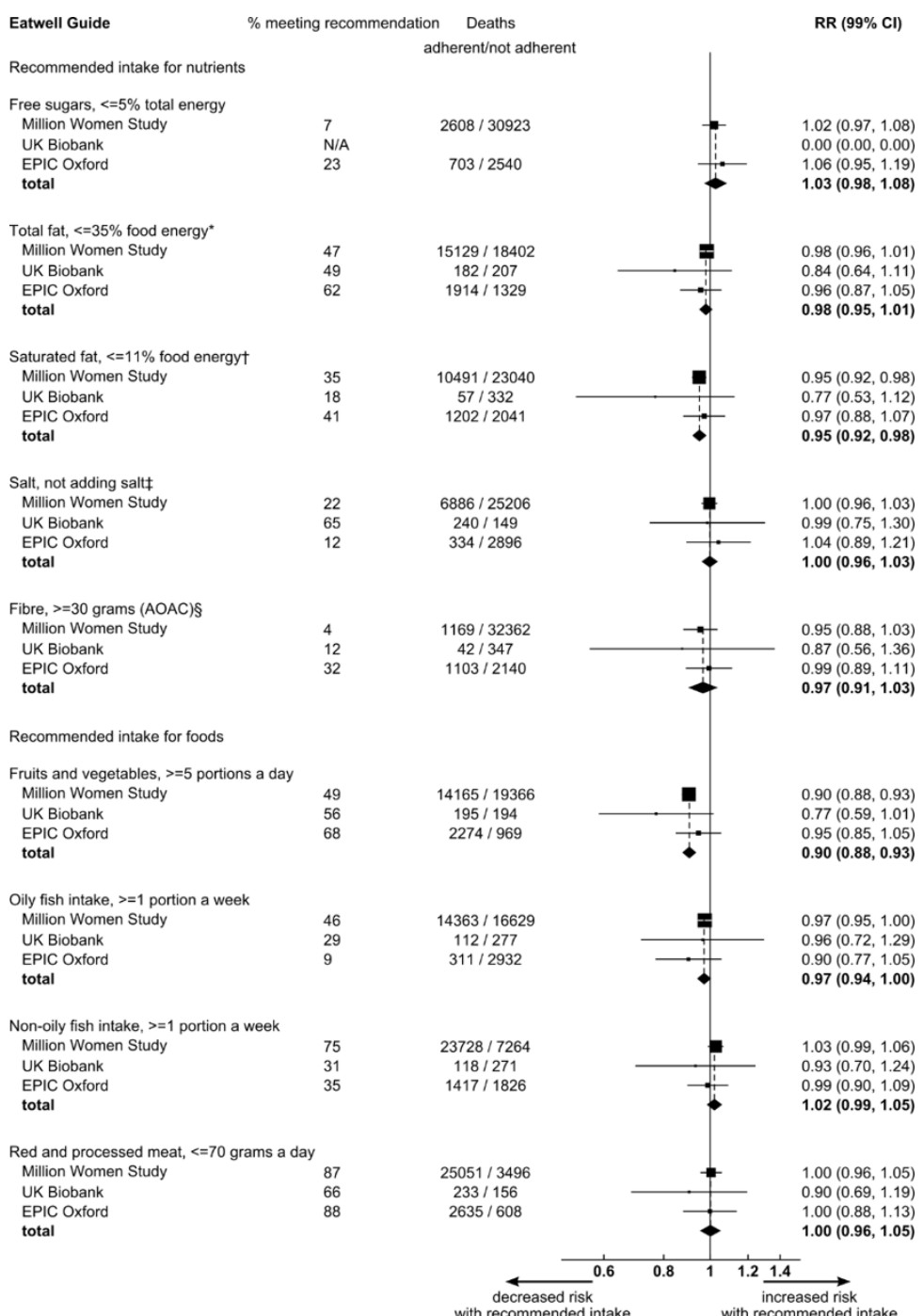

**Figure 3** Mortality risk ratios for the association between adhering to specific Eatwell Guide recommendations and total mortality. *Recommendation was based on food energy and was, therefore, adapted to ≥47% of total energy. *Adapted to ≤33% of total energy. †Adapted to ≤10% of total energy. ‡Information on salt intake was ascertained from the variable 'never adding salt to food at the table or cooking' in the Million Women Study and in the European Prospective Investigation into Cancer-Oxford (EPIC Oxford) study; and from the variable 'not reporting having added salt to food (excluding during cooking)' in any of the online dietary questionnaires included in the UK Biobank. §Fibre intake in the study was determined using the Englyst method (AOAC = Association Of Analytical Chemists) and the recommendation was, therefore, adapted to ≥22.6 g/dL of Englyst fibre. RR, risk ratio.

to other EWG recommendations (figure 3 and online supplementary appendix 4—with recommendation based on dietary reference values for food energy and nutrients for the UK[34]).

## Environmental footprints of diets

Individuals with intermediate-to-high adherence to EWG recommendations showed a reduction in average dietary GHG footprints—compared with those with low and very low EWG adherence—of 12% and 30%, respectively: an average of 3.8 kg $CO_2$eq/day (95% CI: 3.7 to

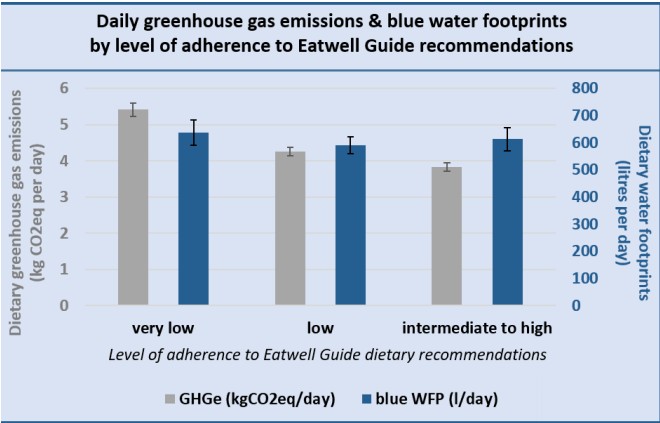

**Figure 4** Average daily GHGe in kg CO$_2$eq and average daily dietary water footprints comparing diets with very low (score 0–2), low (score 3–4) and intermediate-to-high adherence (score 5–9) to the Eatwell Guide dietary guidelines. GHGe, greenhouse gas emissions. WFP, water footprint

3.9 kg CO$_2$eq/day), (4.3 kg CO$_2$eq/day (95% CI: 4.1 to 4.4 kg CO$_2$eq/day) and 5.4 kg CO$_2$eq/day (95% CI: 5.2 to 5.6 kg CO$_2$eq/day) for intermediate to high (score 5–9), low (score 3–4) and very low (score 0–2) EWG adherence, respectively) Dietary blue WFs were similar across adherence groups (figure 4): 637 kg CO$_2$eq/day (95% CI: 590 to 683), 590 kg CO$_2$eq/day (95% CI: 558 to 622) and 612 kg CO$_2$eq/day (95% CI: 571 to 654), respectively, for very low, low and intermediate-to-high adherence to the EWG recommendations. GHGe and WFs changed marginally when adjusting for dietary energy intake (see online supplementary appendix 9).

Mean difference in consumption (in grams/day) of foods between EWG adherent and non-adherent individuals was large (table 1). Associated differences in dietary GHGe were small for fruit and vegetables, oily fish and non-oily fish consumption, and adherence to the recommendation on red and processed meat was associated with lower GHGe (−1.48 kg CO$_2$eq/day; 95% CI: −1.79 to −1.18) (table 1). Differences in blue WFs were small for oily fish and non-oily fish consumption, and adherence to the fruit and vegetable recommendation was associated with a larger blue WF (+28.5 L/person/day; 95% CI: 17.4 to 39.8), while adherence to the red and processed meat recommendation was associated with a lower blue WF (−22.5 L/person/day; 95% CI: −22.7 to −22.3).

## DISCUSSION

Adherence to the EWG is currently low among the UK population. Our analysis of three large UK cohort studies suggests that greater adherence is associated with population health benefits, and using data from the nationally representative NDNS data, we demonstrate that increased EWG adherence is associated with a lower environmental footprint in terms of GHGe, although not water use. Adherence to some EWG recommendations would increase environmental footprints in some instances. Taken together these findings suggest broad benefits

to public health and the environment of adherence to the EWG and provide evidence to support strengthened national action to improve diets in the UK for the benefit of people and the planet.

Our findings support earlier analyses[24] showing that UK diets fully compliant with the EWG have lower environmental footprints. Previous studies of the sustainability of UK diets have found that considerable cobenefits to environment and health could be achieved by meeting WHO dietary guidelines,[15 35] increasing adherence to the EAT–Lancet diet[7] and following a predominantly plant-based diet.[20 23 36 37] While our analysis confirms that reducing consumption of red and processed meat is paramount for lowering environmental footprints of diets, the analysis suggested that population health benefits would be mainly associated with the recommended consumption of fruit and vegetables.

The estimated 7% reduction in mortality and 30% reduction in emissions (or an average absolute reduction of 0.58 tonne GHGe/person/year) through better adherence to the EWG guidelines are similar in magnitude as compared with other population-level interventions aiming multiple benefits for health and the environment. For example, a study evaluating a future scenario of increased active travel and lower-emission motor vehicles in London estimated a 0.72 tonne reduction in per person GHGe as compared with the business-as-usual scenario, as well as a 10%–19% reduction in years of life lost from ischaemic heart disease.[38] A dietary modelling study from the Netherlands estimated impact on GHGe (4%–11%) from substituting 35 g/dL of meat with vegetables, fruit, nuts, seeds, pasta, rice, couscous or fish.[39]

A major strength of this study is its use of four large, high-quality data sources for the UK. A number of sensitivity analyses were conducted to test the robustness of the findings to different assumptions about the causal relationships between variables, and ranges of environmental footprints were used to construct confidence intervals for those relationships. A further strength is the use of empirical rather than modelled diets for the study. Nevertheless, the analyses also have potential weaknesses, among these was the simplification that all diets that met a certain number of recommendations were equally healthy (or unhealthy) regardless of which recommendations were being met, and the assumption that lower consumption of one food group or nutrient could not be compensated by higher consumption of other foods. Low interindividual variance in diets associated with high adherence to some recommendations combined with relatively low overall intake (for example of red and processed meat) may also have resulted in low power to detect diet–health associations.[40] As for all studies measuring dietary intake, assessment is subject to measurement error. However, in the three datasets considered in this study, dietary intake data were collected using different methods, reducing the likelihood of type I errors across all included studies. Data on GHGe were obtained from diverse sources, which used different methods and time periods. Data on

**Table 1** Mean per capita change in environmental footprints from switching* from non-adherence to adherence to food-based EWG recommendations (*from current level of adherence to adherence by all)

| Metric | Unit | Dietary recommendation | | | | | | | | |
|---|---|---|---|---|---|---|---|---|---|---|
| | | Fruit and vegetables | | Oily fish | | Non-oily fish | | Red and processed meat | |
| | | Meeting recommendation | Not meeting recommendation | Meeting recommendation | Not meeting recommendation | Meeting recommendation | Not meeting recommendation | Meeting recommendation | Not meeting recommendation |
| Weighted average consumption | g/day (SE) | 561 (6.47) | 218 (2.00) | 40.3 (1.23) | 1.14 (0.08) | 39.7 (0.85) | 3.61 (0.13) | 31.8 (0.50) | 113 (1.30) |
| Difference in average consumption | g/day | 343 | | 39.2 | | 36.1 | | −81.2 | |
| Mean difference in GHGe achieved by switching to meeting guideline | kg $CO_2$eq/ day (95% CI) | 0.34 (0.29 to 0.38) | | 0.18 (0.04 to 0.31) | | 0.34 (0.23 to 0.45) | | −1.48 (−1.79 to −1.18) | |
| Mean difference in blue WF achieved by switching to meeting guideline | L/day (95% CI) | 28.5 (17.4 to 39.8) | | 10.0 (9.37 to 10.7) | | 8.23 (7.69 to 8.77) | | −22.5 (−22.7 to −22.3) | |

EWG, Eatwell Guide; GHGe, greenhouse gas emissions; WF, water footprint.

WFs were obtained from a single source, but this source used average crop water requirements and yields from years 1996–2005, and these values may, therefore, have changed by the time of the UK dietary survey ~15 years later, resulting in some inaccuracies of food WFs. We attempted to select data on GHGe from surveys with years corresponding to the years of the NDNS, but this was not always possible, and therefore the same inaccuracies may affect the GHG footprints of the diets. Finally, due to data limitations, it was not possible to assess both health and environmental footprints of diets within single datasets.

The EWG dietary recommendations are associated with better health outcomes and lower GHGe but are substantially different from the 'planetary health diet' recently recommended,[6] particularly in terms of red and processed meat consumption—with a much lower amount, maximum amount of meat recommended in the latter. Our analysis suggests that considerable dietary shifts are required in UK dietary habits to meet the EWG recommendations, and that additional substantial changes would be needed to meet the more stringent planetary health diet recommendations. A major determinant of such shifts will be food prices[41 42] and recent analysis has demonstrated that affordability of such diets may vary substantially.[43] Furthermore, it should be noted that an increasing proportion of plant-based foods for human consumption in the UK is imported from abroad.[44] Therefore, shifts in diets towards such foods, and no change in trading strategy, would further increase reliance on foreign production for resilient supply of plant-based foods. Moreover, an increasingly large proportion of these plant-based food imports originates from countries that are highly vulnerable to climate change (eg, countries that are predicted to be highly water deficient by 2030).[32] Care should be taken to avoid that dietary shifts towards EWG adherence (and hence more plant-based diets) would result in substantial virtual water trade—away from water scarce countries—to supply the UK markets.

A fast-tracked nationwide shift towards adherence to the EWG will provide an essential step towards sustainable and healthy diets in the UK, to be followed by careful considerations on how to further improve sustainability beyond EWG adherence. Health services including family doctors must play an active role in promoting adherence to the EWG recommendations to their patients,[45] and thereby contribute directly to population health and environmental sustainability.

**Acknowledgements** The authors thank the women who have participated in the Million Women Study (MWS) as well as the staff from the participating NHS breast screening centres. We also thank NHS Digital in England and the Information Services Division, NHS Scotland for linkage to data on cancers and deaths, and Public Health England for data based on information collected and quality assured by the Public Health England National Cancer Registration and Analysis Service, for which access was facilitated by the Office for Data Release. Data for this study include information collected and provided by the Office for National Statistics. Those who carried out the original collection and analysis of the data bear no responsibility for their further analysis or interpretation. Data access policies for the MWS are available via the study website (http://www.millionwomenstudy.org/).

We thank all participants in the EPIC-Oxford cohort for their invaluable contribution. This research has been conducted using the UK Biobank (UKB) resource under application number 24494. All bona fide researchers can apply to use the UKB resource for health-related research that is in the public interest (https://www.ukbiobank.ac.uk/register-apply/). We thank all participants, researchers and support staff who make the study possible. We thank Paul Appleby and Aurora Perez-Cornago (Cancer Epidemiology Unit, Nuffield Department of Population Health, University of Oxford) for their valuable contribution to data preparation.

**Contributors** PS: literature search, study design, data analysis, data interpretation and manuscript writing. RG, KP and AK: study design, data analysis, data interpretation and manuscript writing. CA-C and AB: data analysis and commenting on manuscript. TJK, VB and ADD: study design and commenting on manuscript.

**Funding** This work was supported by the Wellcome Trust, Our Planet Our Health Programme (Sustainable and Health Food Systems (grant number 205200/Z/16/Z) and Livestock, Environment and People (grant number 205212/Z/16/Z)); Cancer Research UK (grant numbers C8211/A19170 and C570/A11692) and the UK Medical Research Council (grant numbers MR/M012190/1 and MR/K02700X/1). The views expressed in this publication are those of the authors and not necessarily those of the Public Health England or the Wellcome Trust. The final version of the report and ultimate decision to submit for publication was determined by the authors.

**Competing interests** None declared.

**Patient and public involvement** Patients and/or the public were not involved in the design, or conduct, or reporting, or dissemination plans of this research.

**Patient consent for publication** Not required.

**Provenance and peer review** Not commissioned; externally peer reviewed.

**Data availability statement** All data relevant to the study are included in the article or uploaded as supplementary information. All relevant data to the study are included in the article or uploaded as supplementary information. Raw data from National Diet and Nutrition Survey are available (upon request) from UK Data Service https://beta.ukdataservice.ac.uk/datacatalogue/series/series?id= 2000033. Raw data from UK Biobank (UKB), Million Women Study (MWS) and European Prospective Investigation into Cancer - Oxford (EPIC-Oxford) are made available for selected research requests only. (UKB: https://www.ukbiobank.ac.uk/ principles-of-access/; MWS: http://www.millionwomenstudy.org/files/07112018 Datasharingpolicy.pdf; EPIC-Ox: http://www.epic-oxford.org/data-access-sharing-and-collaboration/).

**ORCID iD**
Pauline Scheelbeek http://orcid.org/0000-0002-6209-2284

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
