## [Reviewer comments · BMJ Open]

ARTICLE DETAILS

TITLE (PROVISIONAL)	HEALTH IMPACTS AND ENVIRONMENTAL FOOTPRINTS OF DIETS THAT MEET THE EATWELL GUIDE RECOMMENDATIONS: ANALYSES OF MULTIPLE UK STUDIES
AUTHORS	Scheelbeek, Pauline; Green, Rosemary; Papier, Keren; Knuppel, Anika; Alae-Carew, Carmelia; Balkwill, Angela; Key, Timothy; Beral, Valerie; Dangour, Alan

VERSION 1 - REVIEW

REVIEWER	Kyle F. Davis University of Delaware, USA
REVIEW RETURNED	14-Apr-2020

GENERAL COMMENTS	In general, the authors have satisfied this reviewer's comments. However, the introduction still requires an expansion describing examples of environmental footprint studies that have examined UK diets (e.g., Hoekstra & Mekonnen, 2016, Environ. Res. Lett.; Behrens et al., 2017 PNAS; Vanham et al., 2018 Nature Sustain.; Scarborough et al., 2014 Climate Change; Green et al., 2015 Climatic Change). Further, the additions to the introduction to highlight the novelty of the study relative to previous research were rather brief. As a reader, this reviewer would like to be walked through what previous work has shown within the UK, what it has missed, and how the current study helps to address that gap. One minor point: It would be good if the authors make a note about the difference between the environmental impacts of dietary GHGs (which are global, with generally the same impact regardless of the place of production) vs the environmental impacts of dietary water footprints (which are highly location-specific and depend on the relative abundance of freshwater in the place of production). In other words, reducing GHGs is universally a benefit, while a reduction in water footprint isn't necessarily good if the sourcing of food One other minor comment: The time period for the water footprint data is 1996-2005.
---

VERSION 1 – AUTHOR RESPONSE

Reviewer: Kyle F. Davis

- "In general, the authors have satisfied this reviewer's comments".

Thank you

- "However, the introduction still requires an expansion describing examples of environmental footprint studies that have examined UK diets (e.g., Hoekstra & Mekonnen, 2016, Environ. Res. Lett.; Behrens et al., 2017 PNAS; Vanham et al., 2018 Nature Sustain.; Scarborough et al., 2014 Climate Change; Green et al., 2015 Climatic Change). Further, the additions to the introduction to highlight the novelty of the study relative to previous research were rather brief. As a reader, this reviewer would like to be walked through what previous work has shown within the UK, what it has missed, and how the current study helps to address that gap".

We thank the reviewer for stressing the importance of a more extended introduction. We have elaborated the introduction (now "Background") and discussed the literature suggested by the reviewer. We also have more clearly stressed the novelty of the current study, and how it helps to address a research & practice gap.

- "One minor point: It would be good if the authors make a note about the difference between the environmental impacts of dietary GHGs (which are global, with generally the same impact regardless of the place of production) vs the environmental impacts of dietary water footprints (which are highly location-specific and depend on the relative abundance of freshwater in the place of production). In other words, reducing GHGs is universally a benefit, while a reduction in water footprint isn't necessarily good if the sourcing of food".

We very much agree with the reviewer, and thank them for this helpful comment. We have added a few lines on this to the Background section of the manuscript.

- "One other minor comment: The time period for the water footprint data is 1996-2005". Thank you for pointing this out. We have revised this accordingly